# Time Series Prediction of Gas Emission in Coal Mining Face Based on Optimized Variational Mode Decomposition and SSA-LSTM

**DOI:** 10.3390/s24196454

**Published:** 2024-10-06

**Authors:** Jingzhao Zhang, Yuxin Cui, Zhenguo Yan, Yuxin Huang, Chenyu Zhang, Jinlong Zhang, Jiantao Guo, Fei Zhao

**Affiliations:** College of Safety Science and Engineering, Xi’an University of Science and Technology, Xi’an 710054, China; 18954603644@163.com (J.Z.); yanzg@xust.edu.cn (Z.Y.); 20120089017@stu.xust.edu.cn (Y.H.); z18234611507@163.com (C.Z.); 22220226186@stu.xust.edu.cn (J.Z.); g18339899103@163.com (J.G.); 13571142857@163.com (F.Z.)

**Keywords:** time series prediction of gas emissions, variational mode decomposition, sparrow search algorithm, long short-term memory

## Abstract

The accurate prediction of gas emissions has important guiding significance for the prevention and control of gas disasters in order to further improve the prediction accuracy of gas emissions in the mining face. According to the absolute gas emission monitoring data of the 1417 working face in a coal mine in Shaanxi Province, a GA-VMD-SSA-LSTM gas emission prediction model (GVSL) based on genetic algorithm (GA)-optimized variational mode decomposition (VMD) and sparrow search algorithm (SSA)-optimized long short-term memory (LSTM) is proposed. Firstly, a VMD evaluation standard for evaluating the amount of decomposition loss is proposed. Under this standard, the GA is used to find the optimal parameters of the VMD. Then, the SSA is used to optimize the key parameters of the LSTM to establish a GVSL prediction model. The model predicts each component and finally superimposes the prediction results for each component to obtain the final gas emission result. The results show that the accuracy of the evaluation indexes of the GVSL model and VMD-LSTM model, as well as the SSA-LSTM model and Gaussian process regression (GPR) model, are compared and analyzed horizontally and vertically under three scenarios with prediction sets of 121,94 and 57 groups. The GVSL model has the best prediction effect, and its fitting degree R2 values are 0.95, 0.96, and 0.99, which confirms the effectiveness of the proposed GVSL model for the time series prediction of gas emission in the mining face.

## 1. Introduction

The accurate prediction of gas emissions has important guiding significance for preventing and controlling gas disasters [1]. Improving the accuracy of gas emission prediction is one of the important research branches of gas disaster prevention [2,3].

The static models established by traditional prediction methods, such as the mine statistics method, fractional source prediction method, and neural network prediction [4], fail to consider that gas emission is a dynamic nonlinear system [5]. With the deepening of the research, it cannot meet the requirement of prediction accuracy. In order to improve the prediction accuracy of the model, many scholars introduced influencing factors of gas emissions (coal seam gas content, coal thickness, layer spacing, etc.). A multi-index gas emission prediction model was established [6,7,8,9]. WANG Yanbin analyzed the factors affecting gas emissions in the working face and then predicted the gas emissions based on the PCA-PSO-ELM model. The results show that the prediction effect of the model is better than that of the random forest and extreme learning machine models. Although this method improves the prediction accuracy, it has two disadvantages: First, most mines cannot provide detailed data such as coal thickness and adjacent layer thickness. Second, most prediction models are unable to forecast for a long duration and on a large scale [10,11,12]. Therefore, many scholars have explored and studied the timing prediction model of gas emissions based on the data on gas emissions [13,14,15].

At present, the timing series prediction model of gas emissions mostly adopts the combination prediction method based on signal decomposition, such as wavelet decomposition, empirical mode decomposition, and variational mode decomposition [16,17,18,19]. Among them, variational mode decomposition has better noise robustness than other signal decomposition methods [20]. Some scholars have verified its superiority in the fields of power load and wind speed forecasting [21,22,23]. However, the following problem remains:

(1) The effect of variational mode decomposition mainly depends on the setting of decomposition number k and quadratic penalty factor α, but its value is often set by experience and lacks selection criteria, so it is difficult to guarantee the decomposition effect [24,25].

(2) The prediction model plays an important role in the prediction of gas emissions. Benefiting from the advantages of abstracting and extracting features from input signals layer by layer to dig out deeper potential rule information, the deep learning model has been gradually applied to the field of timing prediction [26]. As a deep learning model, the LSTM model introduces the concept of a time sequence into the network structure, which provides a good effect on time sequence prediction and has achieved good application effects in the fields of power load prediction and photovoltaic power prediction [27,28]. However, there are few studies in the field of gas emission time sequence prediction.

For the above problems, a GA-VMD-SSA-LSTM (GVSL) gas emission prediction model based on genetic algorithm (GA)-optimized variational mode decomposition (VMD) and sparrow search algorithm (SSA)-optimized long short-term memory (LSTM) is proposed.

## 2. Materials and Methods

### 2.1. Variational Mode Decomposition (VMD)

Different from the EMD, LMD, and EEMD decomposition methods of recursive mode decomposition, VMD is a new non-recursive and adaptive signal decomposition method, which was proposed in 2014 [29]. Thanks to the introduction of the variational model, it effectively avoids the endpoint and modal aliasing effects in the recursive mode decomposition method [30].

Based on the concepts of the Wiener filter, Hilbert transform, signal parsing, mixing, and heterodyne demodulation, the steps of VMD construction are proposed as follows [31]:

(1) In order to evaluate the modal bandwidth, the Hilbert transform is introduced to transform the problem into a constrained variational problem. The equation is shown in (1):(1)$$\begin{array}{l} \underset{\left\{u_{k}\},\{\omega_{k}\right\}}{\min}\left\{\sum_{k}{\left\|\partial_{t}[(\delta (t)+\frac{j}{\pi t})*u_{k}(t)]e^{-j\omega_{k}t}\right\|}_{2}^{2}\right\}\\ s.t.\sum_{k}u_{k}=f(t) \end{array}$$

In the formula, *u_k_* is the set of mode decomposition components, *ω_k_* is the set of center frequencies corresponding to the decomposition components, *k* is the number of VMD decompositions, and *f* (*t*) is the input signal to be decomposed.

(2) By introducing the Lagrange multiplier and quadratic penalty factor, the constraint form is transformed into an unconstrained form. The equation is shown in (2):(2)$$\begin{array}{l} L(\left\{u_{k}\right\},\left\{\omega_{k}\right\},\lambda ):=\alpha \sum_{k}{\left\|\partial_{t}[(\delta (t)+\frac{j}{\pi t})*u_{k}(t)]e^{-jw_{k}t}\right\|}_{2}^{2}\\ +{\left\|f(t)-\sum_{k}u_{k}(t)\right\|}_{2}^{2}+\langle \lambda (t),f(t)-\sum_{k}u_{k}(t)\rangle \end{array}$$
where *L* is the augmented Lagrangian, *λ* is the Lagrangian multiplier, and *α* is the quadratic penalty factor.

(3) The alternating direction method of multipliers (ADMM) is introduced to find the saddle point of the augmented Lagrangian and solve the original minimization problem. The optimal solutions of *u_k_* and *ω_k_* in Equation (1) are obtained, and the calculation process is as follows, as shown in Equations (3) and (4).
(3)$${\hat{u}}_{k}^{n+1}(\omega )=\frac{\hat{f}(\omega )-\sum_{i\neq k}{\hat{u}}_{i}(\omega )+\frac{\hat{\lambda }(\omega )}{2}}{1+2\alpha {\left(\omega -\omega_{k}\right)}^{2}}$$
(4)$$\omega_{k}^{n+1}=\frac{{\int_{0}^{\infty }\omega \left|{\overset{⌢}{u}}_{k}(\omega )\right|}^{2}d\omega }{{\int_{0}^{\infty }\left|{\overset{⌢}{u}}_{k}(\omega )\right|}^{2}d\omega }$$

In the equation: ${\hat{u}}_{k}^{n+1}(\omega )$ is the Wiener filter of the current residual $\hat{f}(\omega )-\sum_{i\neq k}{\hat{u}}_{i}(\omega )+\frac{\hat{\lambda }(\omega )}{2}$, and $\omega_{k}^{n+1}$ is the center of gravity of the current modal power spectrum.

### 2.2. Variational Mode Decomposition Based on Genetic Algorithm Optimization

Among the VMD decomposition parameters, k determines the number of mode decompositions, and α affects the fidelity and effect of the modal components. In the field of time series prediction, the selection of parameter values is mainly based on an empirical setting and spectrum analysis setting. The former lacks a theoretical basis and makes it difficult to ensure decomposition quality, while the latter is limited by the absolute gas emission data collected in this paper, which cannot be analyzed for its spectrum and, thus, makes it difficult to determine the parameter values. Therefore, the GA algorithm was introduced to optimize the parameters of *k* and *α* in VMD to ensure the decomposition effect of absolute gas emission data in VMD.

In an ideal situation, the data reconstructed by the VMD decomposition component is the same as the original data, but there is often a decomposition loss in the actual decomposition. In order to evaluate this part of the loss, the root mean square error (*RMSE*) is introduced, as shown in Equation (5).
(5)$$RMSE=\sqrt{\frac{1}{n}{\sum_{i=1}^{n}\left(y-y^{\prime }\right)}^{2}}$$

In the formula, *n* is the number of samples collected for the absolute gas emission, *y* is the measured value of absolute gas emission, and *y*′ is the reconstructed value of gas emission of the VMD decomposition component.

As a method to measure the complexity of nonlinear and non-stationary signals, the sample entropy has the advantages of no need for self-matching and small error [32]. The entropy value represents the complexity of time series data. Therefore, the sample entropy is introduced to evaluate the VMD decomposition effect. The smaller the sample entropy value, the more obvious the periodicity, the less the noise interference, the lower the complexity of the time series data, and the more conducive it is to the training and learning of the gas emission prediction model.
(6)$$S\mathrm{ampEn}(m,r)=\underset{N\to \infty }{\lim}\left\{-\ln[\frac{A^{m}(r)}{B^{m}(r)}]\right\}$$

In the formula, *m* is the window length of the sequence when calculating the sample entropy, *r* is the similarity tolerance threshold, A*^m^*(*r*) is the probability of two sequences matching *m* + 1 points under the similarity tolerance threshold *r*, and B*^m^*(*r*) is the probability of two sequences matching m points.

The RMSE and sample entropy are fused to construct the fitness function, which is expressed as Equation (7). It can not only reflect the sequence loss information after decomposition but also contain the sequence decomposition effect.
(7)fitness=RMSE·SampEn(m,r)

At this point, the VMD parameter selection problem is transformed into the following constrained optimization problem, and the expression is as follows (8):(8)$$\underset{\alpha ,k}{\min}\left\{RMSE*S\mathrm{ampEn}\right\},s.t.\left\{\begin{array}{c} \alpha \in \left\{200,2000\right\}\\ k\in \left\{3,10\right\} \end{array}\right.$$

Note: *k* and *α* optimization range reference [33].

### 2.3. GA Optimizing VMD

The GA is used to solve the above constraint optimization problem. The steps are as follows:

(1) Input the absolute gas emission time sequence data to be decomposed and set the GA maximum iteration times, population size, crossover probability, and other parameters.

(2) Define the optimization dimension and define its optimization scope.

(3) Initialize the population and generate the initial population. Under the current population, VMD decomposition is performed on the absolute gas emission time series data, and the RMSE and sample entropy of the reconstructed data and the measured data are calculated. The initial best fitness value is calculated, and the initial best chromosome is recorded according to Formula (7).

(4) Iterative optimization is performed according to the maximum number of iterations, and the selection, crossover, and mutation operations are performed to calculate the fitness values of various populations and their respective chromosomes in each iteration.

(5) According to Formula (8), the optimal chromosome is selected and decoded to obtain the optimal values of α and k.

(6) The value of k in the VMD decomposition parameter is a positive integer. The round(k) rounding process is carried out to obtain the final VMD parameter optimization value.

### 2.4. Sparrow Search Algorithm to Optimize Long and Short-Term Memory Networks’ Long Short-Term Memories (LSTM)s

LSTM is a type of deep learning. It solves the problems of gradient explosion and gradient disappearance in RNN training through the “gate” structure [34], which can effectively learn long-term dependence and is widely used in the prediction and classification of time series data. Figure 1 shows the unit structure of LSTM.

Its unit structure realizes information protection and control by forgetting the gate, input gate, and output gate. The LSTM steps are as follows:

(1) Information discard: The cell output at time *t* − 1 and cell input at time *t* are read, and Equation (9) is used to complete the information discarded at time *t*.
(9)$$f_{t}=\sigma (W_{f}\cdot [h_{t-1},x_{t}]+b_{f})$$

(2) Information update: Equation (10) determines which information needs to be updated by the sigmoid layer, Equation (11) determines how much new information is added to time t by the tanh layer, and the last two parts are combined by Equation (12) to complete the new cell information update.
(10)$$i_{t}=\sigma (W_{i}\cdot [h_{t-1},x_{t}]+b_{i})$$
(11)$$c_{t}^{\prime }=\tanh(W_{c}\cdot [h_{t-1},x_{t}]+b_{c})$$
(12)$$c_{t}=f_{t}\cdot C_{t-1}+i_{t}\cdot c_{t}^{\prime }$$

(3) Information output: Equation (13) is used to determine which part of the cell information needs to be output by the sigmoid layer, the cell state is processed by the tanh layer, and the final information output is completed in conjunction with Equation (14).
(13)$$O_{t}=\sigma (W_{O}\cdot [h_{t-1},x_{t}]+b_{O})$$
(14)$$h_{t}=O_{t}\cdot \tanh(C_{t})$$

In the formula, *h_t_*_−1_ is the output of the previous moment, *x_t_* is the input of the current moment, the sigmoid activation function, tanh is the hyperbolic tangent activation function, *W_f_*, *W_i_*, *W_c_*, and *W_O_* are the weight values of different ‘gates’, and *b_f_*, *b_i_*, *b_c_*, and *b_O_* are the bias values of different ‘gates’.

### 2.5. SSA Optimizing LSTM

The SSA is a new swarm intelligence optimization algorithm proposed by Xue [35], which is superior to GWO, BA, and other swarm intelligence optimization algorithms in convergence speed, robustness, and stability [36]. In order to avoid over-fitting or under-fitting of the LSTM model caused by human experience, the SSA is introduced to optimize hyperparameters such as MaxEpochs and InitialLearnRate in LSTM so as to establish the optimal gas emission prediction model. The SSA optimization LSTM steps are as follows:

(1) Input the VMD decomposition time sequence data and set the maximum number of iterations of the SSA, population number, security warning value, and other parameters.

(2) Define the optimization dimension and define its optimization scope.

(3) The population is initialized, and the fitness value corresponding to each sparrow is calculated according to Equation (15) and sorted. The initial global optimal fitness value is determined according to the sorting result, and the initial global optimal position is recorded.
(15)$$fitness=MSE=\frac{1}{n}\sum_{i=1}^{n}\left(IMF-IMF^{\prime }\right)$$

In the formula, *n* is the number of *IMF* samples, *IMF* is the modal data of the VMD decomposition, and *IMF*′ is the prediction data of the LSTM model.

(4) Iterative optimization is performed according to the maximum number of iterations, and the fitness values corresponding to each sparrow under each iteration are calculated and their positions recorded.

(5) Optimize the best fitness value and the best position according to Equation (16), and the best position obtained is the final optimization value of each parameter.
(16)$$\min\left\{MSE\right\},s.t.\left\{\begin{array}{c} \mathrm{numHiddenUnits}\in \left\{2,200\right\}\\ \mathrm{InitialLearnRate}\in \left\{0.0001,1\right\}\\ L2\mathrm{Regularization}\in \left\{0.00001,1\right\}\\ \mathrm{MaxEpochs}\in \left\{2,300\right\} \end{array}\right.$$

### 2.6. Construction of GA-VMD-LSTM Prediction Model

Based on the above analysis, the process of the prediction model in Figure 2 is constructed. The specific steps are as follows:

(1) Pre-processing of the absolute gas emission time series data. Detect missing values and outliers to ensure data integrity.

(2) VMD data decomposition. Firstly, the new fitness was constructed as the evaluation standard, and then the GA optimization algorithm was used to optimize the VMD parameters k and α to obtain the optimal VMD parameter settings. In this way, the timing data are decomposed into IMF1, IMF2, …, IMFk.

(3) SSA-LSTM model prediction. The IMFk decomposition data are divided into a training set and a prediction set. The training set data are used to build the SSA-LSTM model, and the prediction set data are used to predict the SSA-LSTM model and output the predicted value.

(4) According to the principle of equal weight superposition, the predicted values of each model were superimposed to obtain the final prediction results of the gas emission.

(5) Model effect evaluation. The mean absolute error (*MAE*), mean absolute percentage error (*MAPE*), root mean square error (*RMSE*), and decision coefficient (R2) were used to evaluate the effect of the prediction model. The equation is from (17) to (20):(17)$$MAE=\frac{1}{n}\sum_{i=1}^{n}\left|\alpha_{t}-{\hat{\alpha }}_{t}\right|$$
(18)$$MAPE=\frac{1}{n}\sum_{i=1}^{n}\left|\frac{\alpha_{t}-{\hat{\alpha }}_{t}}{\alpha_{t}}\right|$$
(19)$$RMSE=\sqrt{\frac{1}{n}\sum_{i=1}^{n}{\left(\alpha_{t}-\hat{\alpha_{t}}\right)}^{2}}$$
(20)$$R^{2}=1-{\frac{\sum_{i=1}^{n}\left(\alpha_{i}-\hat{\alpha_{i}}\right)}{{\sum_{i=1}^{n}\left(\alpha_{i}-\bar{\alpha_{i}}\right)}^{2}}}^{2}$$

In the equation, *α_i_* is the measured data of the gas emission, *α_i_* is the predicted data of the gas emission, and *n* is the number of samples collected.

## 3. Results and Discussion

Taking the 1417 fully mechanized mining face of a coal mine in Shaanxi as the research object, the main coal seam of the 1417 working face is a 4-2 coal seam; the thickness of the coal seam is 4.0~19.0 m, and the average thickness is 10.0 m. The working face adopts the gas control measures of “pre-mining strata hole pre-extraction + roof directional long drilling + upper corner buried pipe extraction + air exhaust”.

The absolute gas emission data of the 1417 working face, from 26 January 2022 to 30 April 2022, were collected, as shown in Table 1.

The outliers and missing values should be detected before constructing the time series prediction model. Data points other than ±1.5 IRQ (the IQR represents the interquartile distance) were taken as outliers to draw a box plot in Figure 3.

It can be seen from Figure 3 that there are seven outliers in the collected data, and the specific values are shown in Table 2.

For missing data, through the statement shuju [!complete.cases (shuju),], data were detected and no missing values were found.

### 3.1. Data INTERPOLATION

The Table 2 outliers were deleted and interpolated. In order to obtain the best filling method, the outflow data without outliers (99 groups of data from 21 February 2022 to 25 March 2022) were extracted from the original data for random missing [37] processing. The EM interpolation, mean interpolation, linear interpolation, and random forest interpolation were used for interpolation processing, and their mean square errors were compared to optimize the best interpolation method. The mean square error of each interpolation method is shown in Table 3.

As can be seen in Table 3, linear interpolation has the highest interpolation accuracy under six types of miss rates, so linear interpolation is selected for interpolation. The linear interpolation data are shown in Table 4.

### 3.2. VMD Decomposition of Gas Emission Data

The time series data of the absolute gas emission at the 1417 working face after linear interpolation, from 26 January 2022 to 30 April 2022, totaled 283 sets of sample data. The timing diagram is shown in Figure 4

Firstly, the GA algorithm was used to optimize the VMD parameters. The GA-related parameter settings are as follows: the maximum number of iterations is 10; population size 10; crossover probability 0.8; variation probability 0.1; k optimization range [3, 10]; α optimization range [200, 2000]. Its iterative optimization curve is shown in Figure 5.

As can be seen in Figure 5, at the tenth iteration, the minimum fitness value of 0.0096 is obtained, k is 10, and α is 483.70. The VMD decomposition results are shown in Figure 6.

As can be seen in Figure 6, 10 IMF decomposition components of different frequencies were obtained after the VMD decomposition of the absolute gas emission time series data. The IMF1 component, which characterizes the trend change for the absolute gas emission, and the IMF2~IMF10 components with certain periodic characteristics are obtained by decomposition, which reduces the complexity of the original data. In order to evaluate the optimal VMD decomposition effect after GA optimization (k = 10, α = 483.70), the decomposition results of k = 3, 5, 8 (α = 2000) were compared with those of k = 10, and the comparison results are shown in Figure 7 and Table 5.

According to Figure 7, when k = 10, the coincidence between the reconstructed value curve of VMD and the actual value curve of the absolute gas emission is the best. At the sudden change point of gas emission, it is also the closest to the original data. While reducing the complexity of the data, the fluctuation information of the original data is retained. The RMSE values of k = 3, 5, 8, and 10 calculated from Table 5 are 0.87, 0.66, 0.46, and 0.13, respectively, and the data decomposition loss of k = 10 is the lowest. The VMD decomposition effect optimized by the GA is superior to the VMD decomposition effect set by the empirical value in decomposition loss and mutation data retention.

### 3.3. Prediction of Gas Emission

In order to verify the effectiveness of the VMD decomposition algorithm, the decomposed data were divided into training sets and prediction sets in a 4:1 ratio.

The training set data realizes the optimization of the key parameters of LSTM by the SSA, and the optimization value is shown in the table.

The optimal LSTM model parameters were determined by the SSA optimization values in Table 6, thus completing the construction of the GVSL prediction model.

In order to verify the prediction effect of the model, the GVSL prediction model is used to predict the absolute gas emissions of 57 groups in the future of the prediction set. The prediction results are shown in Figure 8, and the prediction errors are shown in Table 7.

It can be seen in Figure 8 and Table 7 that the fitting degree of the predicted value curve and the actual value curve of each component of the GVSL prediction model is high. The average absolute error of prediction of each model fluctuated in the range of 0.0047~0.0460 m^3^/min and was maintained at a low level. The GVSL prediction model of each component has a better prediction effect and successfully predicts the changing trend of each VMD decomposition component.

The prediction results of each component of the GVSL model were superimposed to obtain the final prediction results of absolute gas emission, as shown in Figure 9.

It can be seen from Figure 9 that the reconstructed value curve obtained by each GVSL model coincides with the actual value curve of absolute gas emission. The absolute error ranges from 0.0014 to 0.4895 m^3^/min, and the average absolute error is 0.1156 m^3^/min. The relative error ranges from 0.01% to 2.46%, and the average absolute error is 0.73%. The model can well predict the trend of absolute gas emission in the prediction of the 57 sets.

### 3.4. Comparative Analysis of Prediction Models

In the three scenarios (scenario 1: the sample size of the training set was 162, and the sample size of the prediction set was 121; scenario 2: the sample size of the training set was 189 groups, and that of the prediction set was 94 groups; scenario 3: the training set sample size was 226, and the prediction set sample size was 57), we compared and analyzed the prediction effects of the GVSL, VMD-LSTM, SSA-LSTM, and GPR models. The results are shown in Figure 10 and Table 8.

It can be seen in Figure 10 and Table 8 that the GVSL model has the best prediction effect, especially at the sudden change point of gas emission. It can be clearly seen that the GVSL model is superior to other models, which verifies the feasibility of the model for predicting the absolute gas emission of the working face. The MAE, MAPE, RMSE, and R2 values of the GVSL, VMD-LSTM, SSA-LSTM, and GPR models are compared horizontally. It is concluded that the prediction effect of the GVSL model is better than that of the other three models.

The MAE, MAPE, RMSE, and R2 values of the GVSL model in scenario 1, scenario 2, and scenario 3 were compared, and the R2 values were 0.95, 0.96, and 0.99, respectively. The R2 of the GVSL model in scenario 3 is 0.99, which is closer to 1; that is, the larger the proportion of the training set and prediction set, the more advantages the GVSL model has.

## 4. Conclusions

(1) The interpolation accuracy of random forest interpolation, mean interpolation, EM interpolation, and linear interpolation is compared and analyzed under six types of missing rates. It is determined that linear interpolation is used to interpolate the outliers to ensure the integrity of the data structure.

(2) The optimal k value and α value of VMD under the new fitness function are 10 and 483.70. Through comparative analysis of the VMD decomposition results when k = 3, 5, 8, and 10, it was found that the data decomposition loss of k = 10 is 0.13, which is lower than the pair ratio, and the VMD decomposition effect after GA optimization is the best.

(3) The prediction effects of the GVSL, VMD-LSTM, SSA-LSTM, and GPR models were compared and analyzed under three scenarios. The results show that the prediction accuracy of the GVSL model is the highest, which proves that the model can be effectively applied to the prediction of gas emission in the coal face.

## Figures and Tables

**Figure 1 sensors-24-06454-f001:**
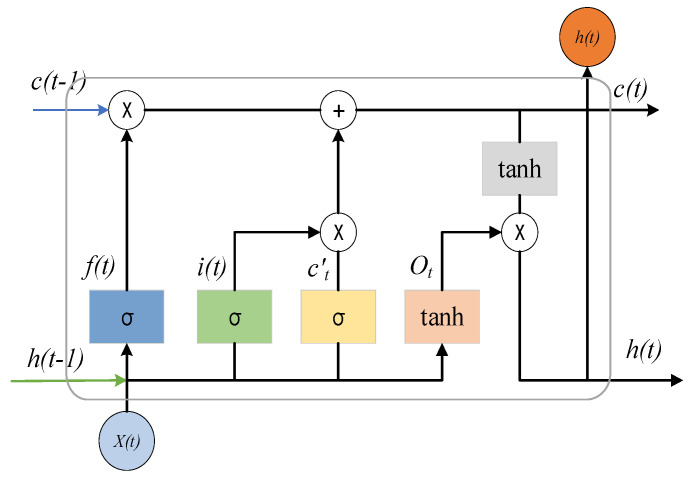
LSTM cell structure.

**Figure 2 sensors-24-06454-f002:**
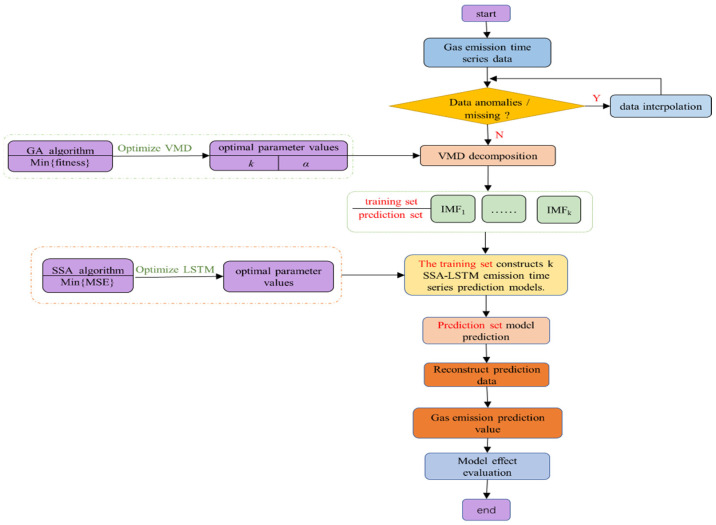
Flow diagram of gas emission prediction model.

**Figure 3 sensors-24-06454-f003:**
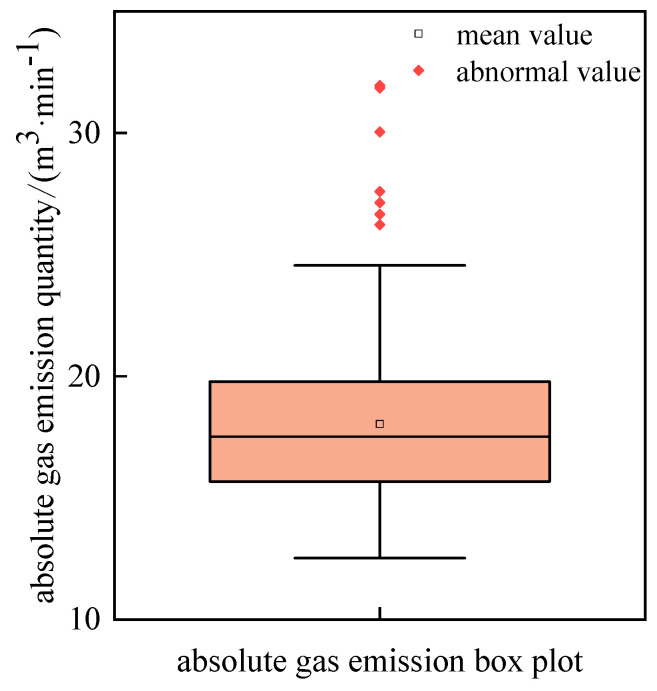
Outlier discriminant boxplot.

**Figure 4 sensors-24-06454-f004:**
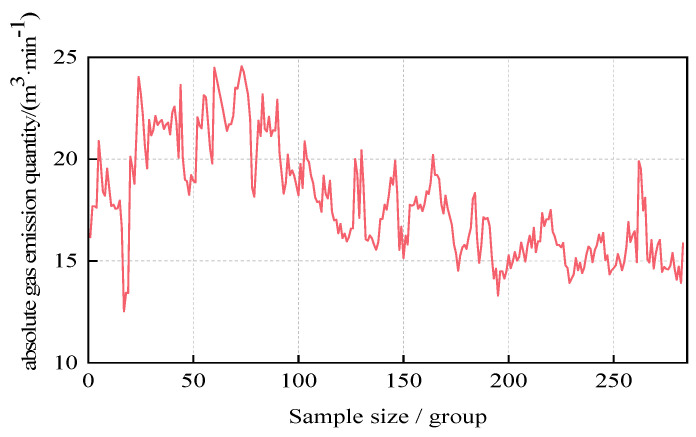
The 1417 mining working face timing diagram.

**Figure 5 sensors-24-06454-f005:**
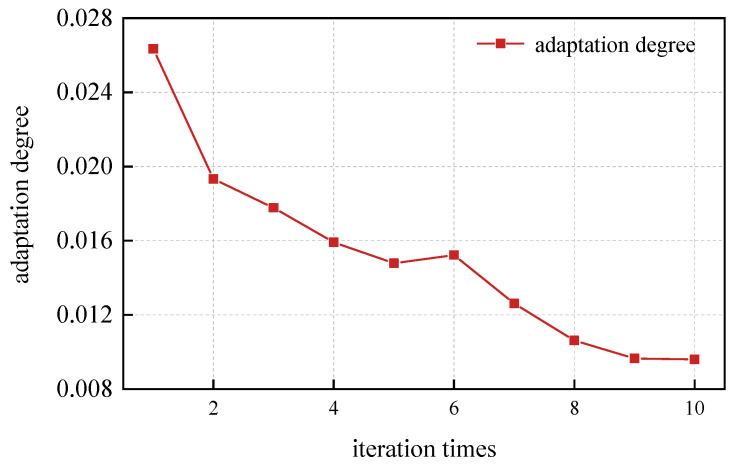
Iterative optimization curve of GA to optimize VMD.

**Figure 6 sensors-24-06454-f006:**
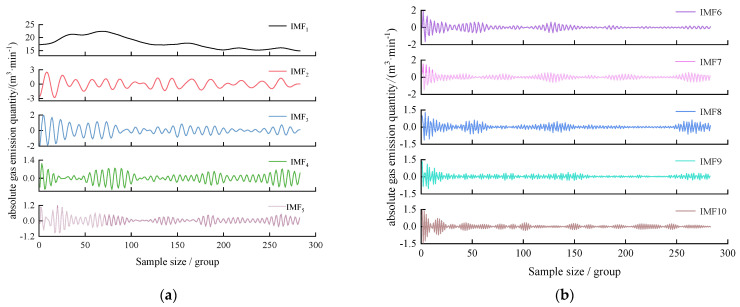
Gas emission of the No. 1417 mining working face by VMD (**a**) IMF1~IMF5 (**b**) IMF6~IMF10.

**Figure 7 sensors-24-06454-f007:**
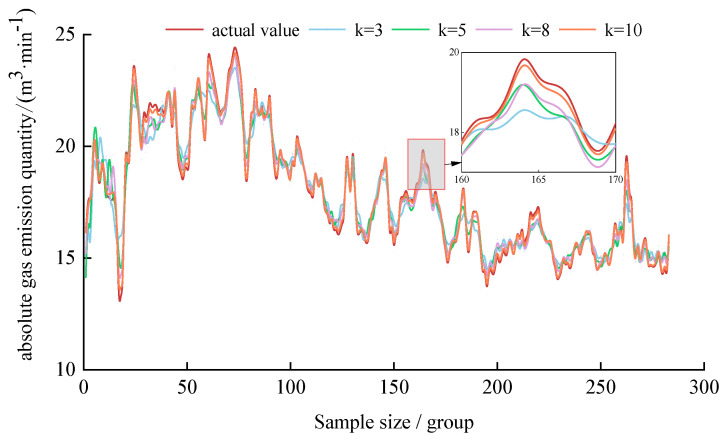
Reconstruction curve by VMD.

**Figure 8 sensors-24-06454-f008:**
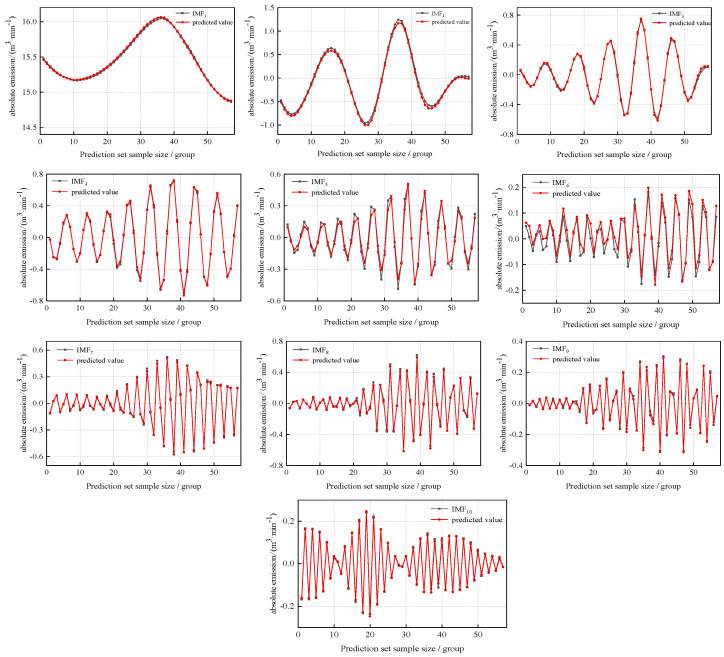
The prediction results of each decomposition component by GVSL.

**Figure 9 sensors-24-06454-f009:**
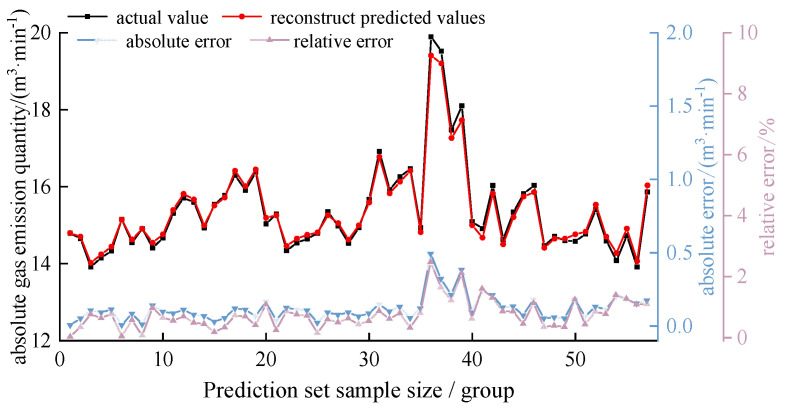
Reconstructed predictions by GVSL model.

**Figure 10 sensors-24-06454-f010:**
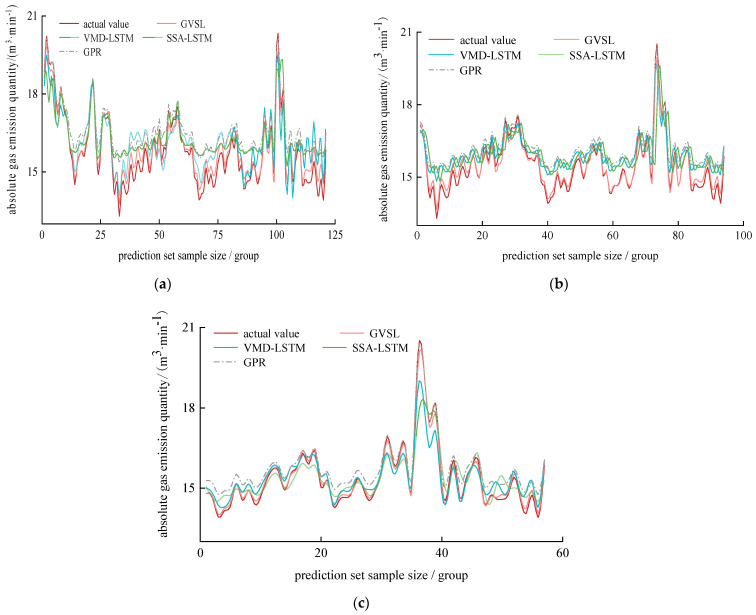
Comparison of different prediction models. (**a**) scenario 1 (**b**) scenario 2 (**c**) scenario 3.

**Table 1 sensors-24-06454-t001:** Gas emission data.

Serial Number	Time	Class	Gas Ventilation Volume/(m^3^·min^−1^)	Gas Drainage Volume/(m^3^·min^−1^)	Absolute Gas Emission Quantity/(m^3^·min^−1^)
1	1/26	16	5.25	10.91	16.16
2	1/27	0	5.51	12.17	17.68
3	1/27	8	4.99	12.70	17.69
4	1/27	16	5.51	12.10	17.61
5	1/28	0	6.30	14.59	20.89
……	……		……	……	……
280	4/29	8	3.15	11.44	14.59
281	4/29	16	3.15	10.93	14.08
282	4/30	0	3.38	11.35	14.73
284	4/30	8	2.93	10.99	13.92
283	4/30	16	4.05	11.81	15.86

**Table 2 sensors-24-06454-t002:** Outliers of gas emission data.

Serial Number	Time	Class	Absolute Gas Emission Quantity/(m^3^·min^−1^)
23	2/3	0	23.22
61	2/15	16	26.66
62	2/16	0	27.14
64	2/16	16	31.95
65	2/17	0	31.85
72	2/19	8	27.59
75	2/20	8	30.04

**Table 3 sensors-24-06454-t003:** Imputation error comparison for random missing.

Mean Square Error of Different Interpolation Methods				
Absence Rate/%	EM Algorithm Imputation	Mean Imputation	linear Interpolation	Random Forest Imputation
5	2.30	3.04	0.11	3.47
10	1.28	1.78	0.13	1.81
15	1.13	1.42	0.16	1.48
sor	1.53	1.66	0.39	1.84
25	1.48	1.81	0.41	2.00
30	1.62	2.20	0.39	2.14

**Table 4 sensors-24-06454-t004:** Linear interpolation fill data.

Serial Number	Time	Class	Absolute Gas Emission Quantity/(m^3^·min^−1)^
23	2/3	0	21.41
61	2/15	16	23.99
62	2/16	0	23.50
64	2/16	16	22.46
65	2/17	0	21.92
72	2/19	8	24.00
75	2/20	8	23.73

**Table 5 sensors-24-06454-t005:** Reconstruction data by VMD.

Serial Number	Actual Value (m^3^·min^−1^)	VMD Decomposition Reconstruction Value (m^3^·min^−1^)			
		k = 3	k = 5	k = 8	k = 10
1	16.16	16.16	14.13	15.18	16.10
2	17.68	17.68	17.27	17.26	17.52
3	17.69	17.69	15.60	17.61	17.56
4	17.61	17.61	18.91	18.48	17.87
5	20.89	20.89	20.78	20.64	20.87
……	……	……	……	……	……
280	14.59	14.69	14.92	14.79	14.66
281	14.08	15.23	14.85	14.77	14.17
282	14.73	15.29	15.45	15.28	14.87
284	13.92	14.64	14.72	14.32	14.07
283	15.86	15.37	15.53	16.05	16.04

**Table 6 sensors-24-06454-t006:** LSTM parameter values for SSA optimization.

Decomposed Component	Num Hidden Units	Max Epochs	InitialLearnRate	L2 Regularization
IMF1	161	255	0.0319	0.0284
IMF2	98	54	0.0631	0.0604
IMF3	200	81	0.0081	0.0001
IMF4	21	16	0.7578	0.8235
IMF5	36	30	0.1323	0.1069
IMF6	115	25	0.0118	0.0001
IMF7	30	73	0.0001	0.0001
IMF8	12	25	0.0001	0.0001
IMF9	6	9	0.0011	0.0010
IMF10	200	60	0.0116	0.0001

**Table 7 sensors-24-06454-t007:** Prediction error of each GVSL model.

GVSL Forecasting Model	Absolute Error (m^3^·min^−1^)		
	Minimum Value	Maximum Value	Mean Value
IMF1	0.0009	0.0266	0.0152
IMF2	0.0178	0.0808	0.0460
IMF3	0.0003	0.0426	0.0146
IMF4	0.0002	0.0648	0.0140
IMF5	0	0.0999	0.0348
IMF6	0	0.0583	0.0218
IMF7	0.0005	0.0307	0.0109
IMF8	0.0006	0.0399	0.0144
IMF9	0.0002	0.0254	0.0083
IMF10	0.0010	0.0189	0.0047

Note: To avoid multiple zeros in two decimal places, four decimal places are reserved here.

**Table 8 sensors-24-06454-t008:** Model evaluation index comparison.

	Evaluating Indicator	GVSL	VMD-LSTM	SSA-LSTM	GPR
Scenario one	MAE	0.27	0.60	0.65	0.77
	MAPE/%	1.72	3.82	4.27	5.14
	RMSE	0.31	0.72	0.83	0.88
	R2	0.95	0.74	0.67	0.68
Scenario two	MAE	0.18	0.52	0.73	0.68
	MAPE/%	1.16	3.50	4.76	4.56
	RMSE	0.22	0.61	0.97	0.77
	R2	0.96	0.74	0.34	0.65
Scenario three	MAE	0.11	0.30	0.37	0.39
	MAPE/%	0.71	1.91	2.33	2.62
	RMSE	0.14	0.41	0.53	0.44
	R2	0.99	0.88	0.80	0.87

## Data Availability

The data presented in this study are available upon request from the corresponding author due to privacy.

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
