# Peer review of "Time Series Prediction of Gas Emission in Coal Mining Face Based on Optimized Variational Mode Decomposition and SSA-LSTM"

_sensors, 2024, doi:10.3390/s24196454_

Round 1

Reviewer 1 Report

Comments and Suggestions for Authors

A model of long short-term memory network optimized by genetic algorithm combined with variational mode decomposition and sparrow search algorithm is proposed to address the problems of low accuracy and imbalanced sample in gas prediction results, for predicting gas emission in coal mine working faces. This combination of methods has certain novelty and can effectively improve prediction accuracy. This research work has certain academic value, but there are still some minor issues that the author needs to address, as follows:

1) The research conducted in the article was not explained in the introduction. Please provide additional clarification.

2) The expression in Figure 2 is unclear, please rephrase.

3) A large number of algorithm abbreviations are used in this article. For ease of reading and understanding, please use the full name where abbreviations first appear in each chapter.

4) Please explain the parameters set when comparing the performance of each model.

5) In the introduction, the innovation of this study compared to existing work should be more clearly explained.

Comments on the Quality of English Language

There are errors in the English writing and it is recommended to check and revise it.

Author Response

  1. The research conducted in the article was not explained in the introduction. Please provide additional clarification.

Response: Dear reviewer, thank you for your valuable comments. We have added explanations for the research section in the paper, which is very helpful for the overall structure of the paper. Thank you again for your valuable suggestion.

  1. The expression in Figure 2 is unclear, please rephrase.

Response: Dear reviewer, thank you for your valuable feedback. We have rephrased Figure 2.

  1. A large number of algorithm abbreviations are used in this article. For ease of reading and understanding, please use the full name where abbreviations first appear in each chapter.

Response: Dear reviewer, thank you for your valuable feedback. We use full names to label the algorithms that first appear in each chapter of the article.

  1. Please explain the parameters set when comparing the performance of each model.

Response: Dear reviewer, thank you for your valuable comments.

  1. In the introduction, the innovation of this study compared to existing work should be more clearly explained.

Response: Dear reviewer, thank you for your valuable comments. We fully agree that traditional gas prediction models currently cannot accurately filter out useless data during the prediction process, which in turn affects the prediction results. In response to the above issues, the variational mode decomposition method in signal decomposition is introduced to decompose and process the gas concentration time series data in advance, and then input it into the prediction model for prediction. However, the effectiveness of variational mode decomposition mainly depends on the number of decompositions k and the setting of the quadratic penalty factor α, which often rely on empirical settings and lack selection criteria, making it difficult to ensure the decomposition effect. This paper proposes a GA-VMD-SSA-LSTM (GVSL) model based on Genetic Algorithm (GA) to optimize Variational Mode Decomposition (VMD) and Sparrow Search Algorithm (SSA) to optimize Long Short Term Memory Network (LSTM). This model optimizes VMD parameters through genetic algorithm to better decompose signals and extract useful features, reducing noise interference. Meanwhile, the LSTM network optimized by SSA can more accurately capture long-term dependencies in time series data, thereby improving the accuracy of predictions.

Reviewer 2 Report

Comments and Suggestions for Authors

To expand the audience of readers, it would be interesting to present the characteristics of the modeling object in the work and reflect the necessary initial data on it in the results.

Author Response

To expand the audience of readers, it would be interesting to present the characteristics of the modeling object in the work and reflect the necessary initial data on it in the results.

Response: Dear reviewer, thank you for your valuable comments. After comparative analysis, the optimal parameters for the model are k=10 and α=483.70. according to figure 7, when k = 10, the coincidence between the reconstructed value curve of VMD and the actual value curve of absolute gas emission is the best. At the sudden change point of gas emission, it is also the closest to the original data. While reducing the complexity of the data, the fluctuation information of the original data is retained. The RMSE values of k = 3,5,8 and 10 calculated from table 5 are 0.87,0.66,0.46 and 0.13, respectively, and the data decomposition loss of k = 10 is the lowest. The VMD decomposition effect optimized by GA is superior to the VMD decomposition effect set by empirical value in decomposition loss and mutation data retention.

Reviewer 3 Report

Comments and Suggestions for Authors

This work presents a study on the prediction of gas emissions in coal mining using a model incorporating GA optimized VMD and SSA optimized LSTM. The model aims to enhance the accuracy of predicting gas emissions, which is crucial for disaster prevention. The study uses absolute gas emission data from a coal mine in Shaanxi Province to validate the model, demonstrating superior predictive accuracy compared to other models through rigorous statistical evaluation. Followings are my concerns:

1. Considering the complex nature of the GA-VMD-SSA-LSTM model, how do the authors assess the practicality of its implementation in real-world mining operations?

2. The manuscript details the use of GA for optimizing VMD parameters and SSA for LSTM. How robust is the model to variations in these parameters?

3. The manuscript would benefit from a more detailed explanation of the underlying mechanisms and the mathematical formulations of the GA-VMD-SSA-LSTM model.

4. More related works could be reviewed such as "Reduction in residential electricity bill and carbon dioxide emission through renewable energy integration using an adaptive feed-forward neural network system and MPPT technique".

5. Could the authors provide a deeper theoretical justification for the choice of SSA and GA in this context?

6. The error metrics used (MAE, MAPE, RMSE, R2) provide a measure of predictive accuracy. Could the authors discuss the choice of these metrics and their relevance to the model's objectives?

Author Response

  1. Considering the complex nature of the GA-VMD-SSA-LSTM model, how do the authors assess the practicality of its implementation in real-world mining operations?

Response: Dear reviewer, thank you for your constructive criticism on the shortcomings in our paper. Evaluating the practicality of the GA-VMD-SSA-LSTM model in practical mining operations requires comprehensive consideration of multiple factors. Through simulation experiments, on-site testing, expert evaluation, and other methods, the performance, computing resources, data acquisition, interpretability, and limitations in practical applications of the model can be comprehensively evaluated to determine whether the model is suitable for actual mining operations and to propose improvement suggestions.

  1. The manuscript details the use of GA for optimizing VMD parameters and SSA for LSTM. How robust is the model to variations in these parameters?

Response: The question you raised is crucial, and the robustness of the model is essential for practical applications.The optimization range of parameters k and α was set during the GA optimization process. The selection of this scope needs to be based on experience and understanding of the problem. If the range is set too small, it may limit the search space of the model, resulting in the inability to find the global optimal solution. If the range is set too large, it may increase computational costs and prolong optimization time. Similar to GA, the population size and iteration times of SSA also affect the optimization results. We need to weigh these two parameters based on the actual problem. The security warning values in SSA are used to control the position updates of sparrows and prevent them from entering dangerous areas. According to the experimental results, the model proposed in this article has good robustness to parameter changes.

  1. The manuscript would benefit from a more detailed explanation of the underlying mechanisms and the mathematical formulations of the GA-VMD-SSA-LSTM model.

Response: Dear reviewer, thank you for your valuable comments.

  1. More related works could be reviewed such as "Reduction in residential electricity bill and carbon dioxide emission through renewable energy integration using an adaptive feed-forward neural network system and MPPT technique".

Response: Dear reviewer, thank you for your valuable comments. We have reviewed it "Reduction in residential electricity bill and carbon dioxide emission through renewable energy integration using an adaptive feed-forward neural network system and MPPT technique"

  1. Could the authors provide a deeper theoretical justification for the choice of SSA and GA in this context?

Response: Dear reviewer, thank you for your constructive criticism on the shortcomings in our paper. Research has shown that SSA has faster convergence speed and stability compared to other optimization algorithms such as GA and PSO. This is particularly beneficial for optimizing hyperparameters of LSTM networks, as parameter optimization of these networks can be very time-consuming. GA can explore the entire search space, which makes it less likely to get stuck in local optima compared to gradient based optimization methods. This is crucial for finding the optimal VMD parameters, as these parameters significantly affect the decomposition accuracy and subsequent prediction performance. SSA performs well in optimizing hyperparameters of LSTM networks, while GA is highly efficient in finding the optimal VMD parameters. This combination utilizes the advantages of both algorithms to achieve a more robust and accurate model. Both algorithms are suitable for different types of optimization problems, making the model framework very flexible and applicable to other time series prediction tasks.

  1. The error metrics used (MAE, MAPE, RMSE, R2) provide a measure of predictive accuracy. Could the authors discuss the choice of these metrics and their relevance to the model's objectives?

Response: Dear reviewer, thank you for your valuable comments. MAE intuitively reflects the average level of prediction error. MAPE represents errors as relative values, making it easier to compare prediction results of different magnitudes and better reflecting the impact of prediction errors on actual values. RMSE provides more comprehensive information on error distribution. R2 intuitively reflects the degree of fit of the model and can be easily compared with other models. Selecting MAE, MAPE, RMSE, and R2 as these four indicators can comprehensively evaluate the predictive performance of the model, including the average level of error, relative error, error distribution, and model fit. By comparing the results of different indicators, we can have a more comprehensive understanding of the advantages and disadvantages of the model and make targeted improvements.
